# The Selectivity for Tumor Cells of Nuclear-Directed Cytotoxic RNases Is Mediated by the Nuclear/Cytoplasmic Distribution of p27^KIP1^

**DOI:** 10.3390/molecules26051319

**Published:** 2021-03-02

**Authors:** Glòria García-Galindo, Jessica Castro, Jesús Matés, Marlon Bravo, Marc Ribó, Maria Vilanova, Antoni Benito

**Affiliations:** 1Laboratori d’Enginyeria de Proteïnes, Departament de Biologia, Facultat de Ciències, Universitat de Girona, Campus de Montilivi, Maria Aurèlia Capmany 40, 17003 Girona, Spain; ggarciagalindo88@gmail.com (G.G.-G.); jessica.castro@udg.edu (J.C.); txusmates.88@gmail.com (J.M.); marlon.bravo@udg.edu (M.B.); marc.ribo@udg.edu (M.R.); 2Institut d’Investigació Biomèdica de Girona Josep Trueta (IdIBGi), 17003 Girona, Spain

**Keywords:** ribonuclease, anticancer drug, drug selectivity, cyclin inhibitors

## Abstract

Although single targeted anti-cancer drugs are envisaged as safer treatments because they do not affect normal cells, cancer is a very complex disease to be eradicated with a single targeted drug. Alternatively, multi-targeted drugs may be more effective and the tumor cells may be less prone to develop drug resistance although these drugs may be less specific for cancer cells. We have previously developed a new strategy to endow human pancreatic ribonuclease with antitumor action by introducing in its sequence a non-classical nuclear localization signal. These engineered proteins cleave multiple species of nuclear RNA promoting apoptosis of tumor cells. Interestingly, these enzymes, on ovarian cancer cells, affect the expression of multiple genes implicated in metabolic and signaling pathways that are critic for the development of cancer. Since most of these targeted pathways are not highly relevant for non-proliferating cells, we envisioned the possibility that nuclear directed-ribonucleases were specific for tumor cells. Here, we show that these enzymes are much more cytotoxic for tumor cells in vitro. Although the mechanism of selectivity of NLSPE5 is not fully understood, herein we show that p27^KIP1^ displays an important role on the higher resistance of non-tumor cells to these ribonucleases.

## 1. Introduction

Members of the pancreatic ribonuclease (RNase) superfamily display an array of biological activities ranging from cytotoxicity to angiogenesis. Of all of them, cytotoxicity is highly attractive since such enzymes can be used, alone or conjugated to ligands or antibodies, as non-mutagenic therapeutic agents for cancer treatment [1,2,3]. Among pancreatic ribonucleases, onconase, an RNase isolated from *Rana pipiens*, reached phase II/III of clinical trials for the treatment of malignant mesothelioma but failed due to its renal toxicity when administered at high concentrations. The generation of cytotoxic variants of human pancreatic RNase (HP-RNase), which has lower renal toxicity, is less immunogenic and displays higher ribonucleolytic activity than onconase [4,5,6], is a potentially useful approach to get non-genotoxic antitumor agents. Among the designed variants of HP-RNase some are cytotoxic because they are either resistant to the cytosolic RNase inhibitor (RI) or because they are targeted to tumor cells through specific ligands that ensure an efficient arrival to the cytosol (for a review see [3]). We anticipated an alternative strategy to produce cytotoxic RNases consistent in their engineering to direct them to the nucleus [7,8], which is free of RI [9].

Nuclear directed-RNases (ND-RNases) are cytotoxic because the changes introduced in their sequence endow them with a non-classical nuclear localization signal (NLS) [10]. They act on the nuclear RNA causing apoptosis of tumor cells [11] through a mechanism that is mediated by the induction of p21^WAF1/CIP1^ and the inactivation of JNK [12]. The characterization of the transcription profiling of ovarian cancer cells treated with ND-RNases showed that these enzymes affect multiple metabolic pathways that are necessary for the development of cancer [13]. Interestingly, ND-RNases are highly cytotoxic for multidrug-resistant (MDR) tumor cell lines and in accordance decrease the expression of the multidrug resistance protein P-gp and are synergistic with doxorubicin [14].

One important feature of any antitumor drug is its selectivity. Here we show that one of the most cytotoxic ND-RNase variant, NLSPE5 [8], produces a lower toxic effect on non-tumor cell lines and, unlike to cancer cell lines, it does not affect their proliferation. The different behavior of NLSPE5 on non-tumor cells is related to its effect on p27^KIP1^ expression and its subcellular localization.

## 2. Results

### 2.1. Selectivity of NLSPE5 In Vitro

We had previously shown that the ND-RNase PE5 is much less cytotoxic for N1 non-tumor cells than for many different tumor cell lines [12]. Here, we have extended the study of the selectivity of the ND-RNases for tumor cells using a much more cytotoxic ND-RNase variant, NLSPE5, tested against a wide panel of tumor and non-tumor cell lines. Onconase was included in the study as a reference. Our results show that NLSPE5 is more selective than onconase in vitro (Table 1, and Appendix A). As it can be seen, the IC_50_ values of NLSPE5 for tumor cells are generally lower than that of onconase and, conversely, its IC_50_ values for normal cells are higher than that of onconase. The ratio between the IC_50_ values of non-tumor and tumor cells ranges from 5 to 18-fold when we compare the two least affected cell lines (NCI-H460/R and CCD-18Co) and the most affected (NCI-H460 and HEK-293) by NLSPE5, respectively. In contrast, these values for onconase only range between 0.25 and 1.8-fold. We foresaw characterizing how this ND-RNase variant affects non-tumor cells.

### 2.2. The Effect of NLSPE5 on Proliferation and Viability Differs between Tumor and Non-Tumor Cells

The MTT assay does not allow discerning whether a drug exerts a cytotoxic or a cytostatic effect [12]. Therefore, in the present study we have investigated the effect of NLSPE5 on the cell growth and viability of the least sensitive non-tumor cell line (CCD-18Co) using the trypan blue assay. For comparison we have carried out the same study on the NCI-H460/R MDR cell line. Three different concentrations of NLSPE5 equal or below the IC_50_ for each cell line were investigated for up to 72 h. Figure 1 shows the effect on the proliferation and viability of both cell lines when incubated with these concentrations of NLSPE5.

When comparing both cell lines, we can see that the effect of NLSPE5 on cell viability is much lower in non-tumor cells, as the same rate of viable cells is achieved using much higher concentrations of the ND-RNase. NLSPE5 reduces the cell proliferation of NCI-H460/R cells even at 24 h of incubation and this effect increases along the time. In contrast, NLSPE5 does not seem to affect the proliferation of CCD-18Co cells, at least at the incubation times investigated, being the decrease of total cells equal to that of viable cells. It is worth mentioning that the effect of NLSPE5 on the NCI-H460/R cell line is very similar to that previously observed for another ND-RNase named PE5 in the ovarian MDR cell line NCI/ADR-RES [12] but for NCI-H460/R cells the effect on proliferation is more evident. In summary, NLSPE5 has a minimal effect on the viability of non-tumor cells and likely does not affect their proliferation whereas on tumor cells, the ND-RNase decreases both their viability and proliferation.

This different behavior of NLSPE5 on the non-tumor and tumor cell lines was confirmed by analyzing the cell cycle phase distribution of both cell lines upon treatment with the ND-RNase (Table 2, and Appendix A). For non-tumor CCD-18Co cells, the cell cycle phase distribution is nearly unaltered even when the cells are treated with a concentration that is two-fold the IC_50_ value for this cell line (11 µM). This result indicates that NLSPE5 does not arrest the cell cycle of CCD-18Co, which is in agreement with the lack of effect of this drug on the proliferation rate of these non-tumor cells. In contrast, treatment of NCI-H460/R cells with NLSPE5 produces a clear unbalance of the cell cycle phase distribution even when only near a half of the IC_50_ concentration value for this cell line is used (0.25 µM). At this concentration, the S and G_2_/M phases increase around 50% their proportion whereas the G_0_/G_1_ phase is clearly reduced indicating that cell proliferation arrest is produced mainly at the S and G_2_/M cell cycle phases. A significant fraction of the cells are on sub-G_1_, which is indicative that these cells have entered on apoptosis. This effect is higher on tumor cells (Table 2).

### 2.3. Analysis of the Mechanism of Cytotoxicity of NLSPE5 on CCD-18Co Cells

In order to check whether NLSPE5 induces the apoptosis of the non-tumor CCD-18Co cells, activation of procaspase-3, -8 and -9 was investigated (Figure 2). NLSPE5 induces the activation of the three procaspases reaching its maximum at 48 h of drug incubation. Caspase activation was higher for the effector caspase-3 than for the initiator caspases-8 and -9.

Additionally, we investigated by Western blot the effect produced by NLSPE5 on the level of different cell cycle- and apoptosis-related proteins in the non-tumor cells. Cellular extracts from CCD-18Co treated for 72 h with 5.5 µM NLSPE5 were subjected to immunoblotting using antibodies against Bcl-2; Bax; XIAP; JNK and its phosphorylated form p46JNK; cyclin D_1_; cyclin E; p27^KIP1^ and p21^WAF1/CIP1^ (Figure 3). The amount of Bcl-2, Bax and XIAP, as well as the phosphorylated and unphosphorylated forms of JNK, in treated cells are nearly equal to those of control cells. A significant increase is observed for p21^WAF1/CIP1^, p27^KIP1^ and cyclin E. p27^KIP1^ is known to display antiapoptotic activity when accumulated in the cytoplasm [15]. Therefore, we investigated by western blot the effect of NLSPE5 on the accumulation of this inhibitor on both, nucleus and cytoplasm, of NCI-H460/R and CCD-18Co cells (Figure 4). Regarding tumor NCI-H460/R cells, the amount of p27^KIP1^ in either nucleus or cytoplasm did not change upon treatment with NLSPE5. The amount of p27^KIP1^ in the nucleus was nearly undetectable in CCD-18Co cells but significantly increased in the cytoplasm when they were treated with NLSPE5. We checked for the presence of histone H3, a nuclear marker, and GAPDH, a cytosol marker, both in the nucleus and cytosol fractions using specific antibodies. Although we observed faint bands of histone H3 and GAPDH in the cytosol and the nucleus, respectively (5% of histone H3 in the cytoplasm and 20% of GAPDH in the nucleus), the differences observed in the levels of p27^KIP1^ between compartments cannot not be caused by cross-contamination (Figure 4C).

## 3. Discussion

Over the last decades many efforts have been made to develop highly specific antitumor drugs. Single-target drugs are considered as safer drugs provided they act specifically on cancer-related processes and therefore not affecting non-tumor cells. In this sense, the understanding of the molecular mechanisms that generate and maintain cancer cells has led to the identification of cancer-specific processes that can be specifically drugged. For instance, chronic myeloid leukemia (CML) is characterized by an up-regulation of Bcr-Abl tyrosine kinase oncogene and CML is successfully treated with imatinib, a drug that specifically targets the protein coded by this oncogene. Although the use of a single drug has proven effective in some cancers, usually these drugs are not valid for other cancers due to the high diversity of molecular pathways triggered in each case. In addition, cancer is a very complex disease to be eradicated with a single targeted-drug. Moreover, a typical cancer harbors between two and eight pathogenic mutations per tumor [16] so targeting a single protein may be suboptimal. Even such a successful drug as imatinib, to which most of the patients respond in the chronic phase of the disease, characterized by a relative genome stability, it is not effective in late disease’s phases when highly dynamic genome instability is present. In addition, drug resistance in cancer emerges very frequently during treatment particularly with targeted therapies designed to inhibit specific molecules [17]. Alternatively, the development of drugs with a multi-target action is an interesting approach that may overcome these concerns even if they lead to a decrease of tumor specificity.

ND-RNases are pleiotropic (multi-targeted) drugs that act on RNA affecting the expression of multiple genes implicated in metabolic and signaling pathways that are critic for cancer maintenance [13]. Since most of these targeted pathways are not highly relevant for non-proliferating cells, we envisioned the possibility that ND-RNases could be specific for tumor cells. We tested the effect of NLSPE5 on different non-tumor cells lines. HaCat cell line is aneuploid but, despite exhibiting phenotypic traits of transformation in vitro, is non tumorigenic, closely resemble normal human keratinocytes in their growth and differentiation potential and expresses a virtually normal pattern of differentiation [18,19,20]. CCD-18Co cells are primary normal human colon fibroblasts and have been extensively used as a healthy non-stem-cell and non-cancer-cell in vitro reference model [21,22,23,24]. MFC10A cells are derived from benign proliferative breast tissue and spontaneously immortalized without defined factors. They are not tumorigenic and do not express estrogen receptor. Furthermore, when cultured on top of Matrigel^TM^, MCF10A cells are capable of forming acinus-like spheroids with a hollow lumen [25]. This structure is covered by basement membrane and formed by polarized and organized cells [26]. Regarding 1BR3G cells, they are SV40 immortalized skin fibroblasts derived from a phenotypically normal individual with a normal capacity for DNA DSB repair. They have also been previously used as normal cells [27,28,29]. Finally, HEK-293 cells were generated by transformation of cultured normal human embryonic kidney cells with sheared human adenovirus type 5 DNA [30]. Although these cells are immortalized by a known oncogene, they are not malignant [31] and have previously been used as a non-tumor cell model [32,33]. Interestingly, among these cell models, the best established as a cell model of normal cells (CCD-18Co) is the less affected by NLSPE5.

We have shown that the cytotoxic effect of NLSPE5 is mediated by caspase activation (Figure 2). The activation of the procaspases in CCD-18Co cells is lower than that previously observed in the MDR cell line NCI/ADR-RES [8]. This result is in accordance with the observation that NLSPE5 produces a higher cytotoxic effect on the tumor cell lines than on CCD-18Co cell line.

p21^WAF1/CIP1^, p27^KIP1^, cyclin D1, and cyclin E are the downstream targets of p53 but we have not found any correlation between the status of p53 and their sensitivity to NLSPE5 among the panel of investigated cell lines. Here, we show that NLSPE5 affects specifically tumor cells in vitro and that the higher resistance of CCD-18Co cells to the RNase-induced apoptosis coincides with the overexpression and cytoplasmic subcellular localization of p27^KIP1^.

NLSPE5 increases the accumulation of cytoplasmic p27^KIP1^ in CCD-18Co cells. This is interesting since we have previously shown that ND-RNases do not alter the expression of this cyclin inhibitor at least in the ovarian cancer cell lines NCI/ADR-RES and OVCAR-8 [8,12,13]. Cytoplasmic accumulation of p27^KIP1^ may explain why non-tumor cells are more resistant to the action of the ND-RNases. It has been previously described that cytosolic p27^KIP1^ prevents cytochrome C release and procaspase 3 activation inhibiting apoptosis [34]. Part of p27^KIP1^ is localized within mitochondria improving its membrane potential and it has been shown that it prevents cardiomyocytes from apoptosis [35]. Finally, it has also been described that p27^KIP1^ protects against inflammatory injury and that in p27^-/-^ mice with glomerulonephritis, apoptosis is increased [36].

We have shown here that the cell cycle phase distribution of CCD-18Co cells is not altered upon its treatment with NLSPE5 even though the expression of some of the cell cycle-regulating proteins is affected. This could be explained by a compensation effect of the overexpression of the cyclins and cyclin inhibitors.

It could be expected that since ND-RNases act on multiple cell targets, they could affect tumor and non-tumor cells equally but we have shown here that this is not the case. Altogether these results show that the ND-RNases do not act on indiscriminate pathways but mainly on those that affect the tumor cells. These results are very promising for the development of new pleiotropic antitumor drugs.

## 4. Materials and Methods

### 4.1. RNase Expression and Purification

NLSPE5 and onconase were produced and purified from *E. coli* BL21 (DE3) cells transformed with the corresponding vector as described previously [8,37,38]. The molecular mass of each variant was confirmed by matrix-assisted laser desorption/ionization time-of-flight mass spectrometry in the “Unitat cientificotècnica de suport” of the Institut de Recerca of the Hospital Universitari Vall d’Hebron (Barcelona, Spain). The protein concentration of each variant was determined using a MODEL spectrophotometer (Perkin Elmer, Walthman, MA, USA) using an extinction coefficient of ε_280_ = 7950 M^−1^ cm^−1^ for NLSPE5 and of ε_280_ = 10,470 M^−1^ cm^−1^ for onconase, calculated using the method devised in reference [39].

### 4.2. Cell Lines and Culture Conditions

HeLa human cervical cancer cell line, HaCaT human keratinocytes, CCD-18Co human colon fibroblasts, and Sk-Br-3 human breast cancer cell line were obtained from Eucellbank (Universitat de Barcelona, Barcelona, Spain); NCI-H460/R human lung cancer MDR cell line was a generous gift from Dr. Sabera Ruzdijić of the “S. Stanković” Institute for Biological Research (Belgrade, Serbia); NCI-H460 human lung cancer cell line and OVCAR-8 human ovarian cancer cell lines were obtained from the National Cancer Institute-Frederick DCTD tumor cell line repository; MCF10A human mammary gland epithelial cells were acquired from the American Type Culture Collection (ATCC, Rockville, MD, USA); and HEK-293 human embryonic kidney cells and 1BR3G human epithelial fibroblasts were obtained from the European Collection of Authenticated Cell Cultures (ECACC, Porton, UK).

Cells were routinely grown at 37 °C in a humidified atmosphere of 5% CO_2_. HeLa, HEK-293, HaCaT, CCD-18Co and 1BR3G cells were cultured in DMEM (Gibco, Berlin, Germany) supplemented with 10% fetal bovine serum (FBS) (Gibco), 50 U/mL penicillin and 50 µg/mL streptomycin (Gibco). Sk-Br-3 cells were grown on McCoy’s medium (Gibco) supplemented with 10% FBS (Gibco), 50 U/mL penicillin and 50 µg/mL streptomycin. NCI-H460, NCI-H460/R and OVCAR-8 cells were grown in RPMI (Gibco) supplemented with 10% FBS (Gibco), 50 U/mL penicillin and 50 µg/mL streptomycin (Gibco). NCI-H460/R cells were maintained in media containing 0.1 µM doxorubicin (Tedec-Meijic Farma, Madrid, Spain). MCF10A cells were grown in DMEM/F-12 (Gibco) supplemented with 5% horse serum (Gibco, Gaithersburg, MD, USA), 100 ng/mL cholera toxin (Sigma Aldrich, Saint Louis, MI, USA), 20 ng/mL human epidermal growth factor (EGF) (Gibco, Gaithersburg, MD, USA), 10 μg/mL insulin (Sigma Aldrich), 0.5 μg/mL hydrocortisone (Sigma Aldrich), 50 U/mL penicillin and 50 μg/mL streptomycin (Gibco). Cells remained free of *Mycoplasma* and were propagated according to established protocols.

### 4.3. Cell Growth Assays

Cells were seeded into 96-well plates at the appropriate density, i.e., 1100 (for HeLa), 1500 (for OVCAR-8), 1900 (for NCI-H460), 2000 (for CCD-18Co), 2500 (for MCF10A and HEK-293), 3000 (for Sk-Br-3, NCI-H460/R and 1BR.3.G), 3500 (for HaCaT). After 24 h of incubation, cells were treated with various concentrations of RNase for 72 h. Drug sensitivity was determined by the MTT method essentially following the manufacturer’s instructions (Sigma) and according to [14]. All data are described as the mean ± standard error (SE) of at least three independent experiments with three replicas in each.

### 4.4. Analysis of Cytostatic and Cytotoxic Effects of NLSPE5

CCD-18Co cells and NCI-H460/R cells (1.43 × 10^5^ and 3.0 × 10^5^ cells/60-mm dish, respectively) were treated with different concentrations of NLSPE5 (0.7–11 µM) for 24, 48 and 72 h. After treatment, attached and floating cells were harvested and washed in cold phosphate buffered saline (PBS). The proliferation rate and viability in control and RNase-treated cultures was estimated by cell count using a hemocytometer combined with the trypan blue exclusion assay.

### 4.5. Cell Cycle Phase Analysis

Cell cycle phase analysis was performed by propidium iodide (PI) staining. CCD-18Co and NCI-H460/R cells (3.6 × 10^5^ and 5.0 × 10^5^ cells/100-mm dish, respectively) were treated with NLSPE5 (11 μM for CCD-18Co and 0.25 μM for NCI-H460/R) for 72 h. Cells were then harvested and fixed with 70% ethanol for at least 1 h at −20 °C. Fixed cells were harvested by centrifugation and washed in cold PBS. These collected cells were resuspended in PBS (1–2 × 10^6^/mL) and treated with RNase A (100 μg/mL) and PI (40 μg/mL) (Molecular Probes, Eugene, OR, USA) at 37 °C for 30 min prior to flow cytometric analysis. A minimum of 10,000 cells within the gated region were analyzed on a FACSCalibur flow cytometer (BD Biosciences, San Jose, CA, USA). Cell cycle distribution was analyzed using the FlowJo software (FlowJo LLC, Ashland, OR, USA).

### 4.6. Procaspase Activation Assay

Caspase-3, -8 and -9 catalytic activities were measured using the APOPCYTO Caspase-3, -8 and -9 colorimetric assay kits (MBL, Nagoya, Japan) following the manufacturer’s protocol. The assay is based on cleavage of the chromogenic substrates, DEVD-pNA, IETD-pNA and LEHD-pNA, by caspases-3, -8 and -9, respectively. Briefly, CCD-18Co cells (6 × 10^5^ cells/100-mm dish) were incubated with 11 µM NLSPE5 for 24, 48 and 72 h in serum-starved medium. Then, attached and floating cells were harvested at 460 xg for 10 min at 4 °C, washed twice in cold PBS, and lysed and centrifuged. The supernatant was recovered, and the protein concentration was determined using the Bradford protein assay (Bio-Rad Laboratories, Hercules, CA, USA). Afterwards, 10 µL of the cell lysate corresponding to 20 µg of total protein, 10 µL of 2× reaction buffer containing 10 mM DTT, and 1 µL of the 10 mM DEVD-pNA, IETD-pNA or LEHD-pNA substrates were mixed. The reactions were then incubated at 37 °C for 4 h. The reaction was measured by changes in absorbance at 405 nm. All data are described as the mean ± SE of three independent determinations.

### 4.7. Western Blot Analysis

CCD-18Co cells (3.6 × 10^5^ cells/100-mm dish) were incubated with 5.5 µM NLSPE5 for 72 h. When necessary, nuclear and cytoplasmic proteins were extracted using the PARIS kit (Applied Biosystems/Ambion, Waltham, MA, USA) according to the manufacturer’s instructions and stored at −80 °C. Quantification using antibodies against ß-actin, Bax, Bcl-2, Cyclin D_1_, Cyclin E, p21^WAF1/CIP1^, GAPDH, p27^KIP1^ (Santa Cruz Biotechnology, Santa Cruz, CA, USA), XIAP (BD Transduction Laboratories, San Jose, CA, USA), and histone H3 (H3K27me3) (Abcam, Cambridge, UK) was performed by western blot as previously described [12]. The linearity of the assay was preliminarily checked for each monoclonal antibody by submitting different amounts of untreated cell extracts to western blotting. All data are described as the mean ± SE of at least three independent determinations.

### 4.8. Statistical Analysis

Statistical analyses were performed with IBM SPSS Statistics 23 software for Windows (version 26; IBM Corp, Armonk, NY, USA). Results were analyzed using the Student’s t test. *p*-values < 0.05 were considered statistically significant.

## Figures and Tables

**Figure 1 molecules-26-01319-f001:**
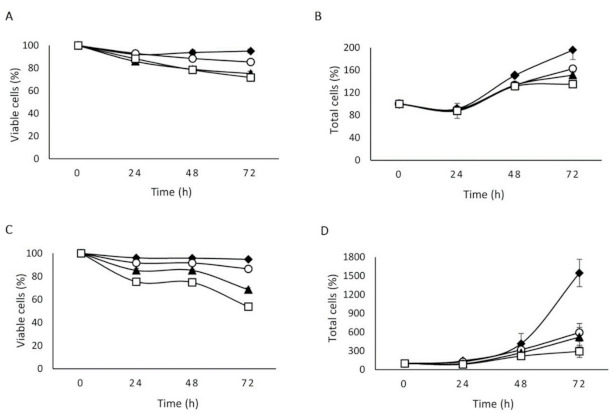
Effect of NLSPE5 on the viability (**A**,**C**) and proliferation (**B**,**D**) of CCD-18Co (**A**,**B**) and NCI-H460/R cells (**C**,**D**) at different incubation times and concentrations (♦ 0 µM; ◯ 0.7 µM; ▲ 3 µM; and □ 5.5 µM for CCD-18Co cells and ♦ 0 µM; ◯ 0.1 µM; ▲ 0.2 µM; and □ 0.4 µM for NCI-H460/R). The plotted points represent means of at least three independent experiments.

**Figure 2 molecules-26-01319-f002:**
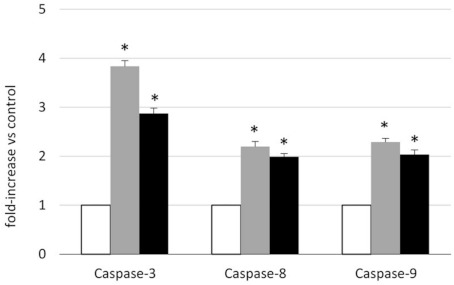
Caspase-3, -8 and -9 activities in CCD-18Co cells treated with NLSPE5. Cells were treated with 11 μM NLSPE5 for 48 (grey bars) and 72 h (black bars). White bars indicate untreated cells (control). Activation of procaspase-3, -8 and -9 was quantified in whole cell lysates using a quantitative colorimetric assay as described in the text. Results are expressed as the mean ± SE of three independent experiments. Differences versus untreated control cells were considered significant at * *p* < 0.05.

**Figure 3 molecules-26-01319-f003:**
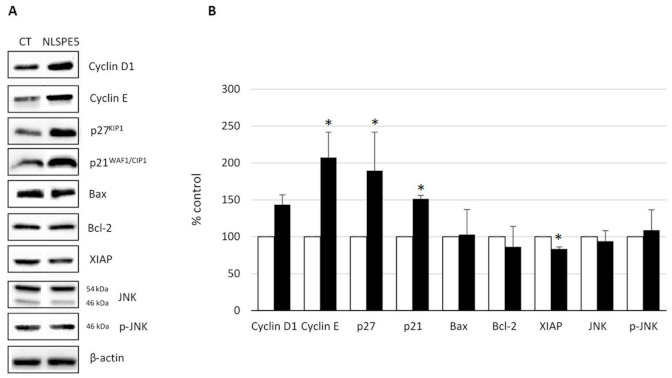
Western blot analysis of the expression of different proteins involved in the control of apoptosis and cell cycle of CCD-18Co. (**A**) Representative Western blots. (**B**) Expression levels relative to that of actin. Values of untreated cells are considered as 100%. Differences versus untreated control cells were considered significant at * *p* < 0.05.

**Figure 4 molecules-26-01319-f004:**
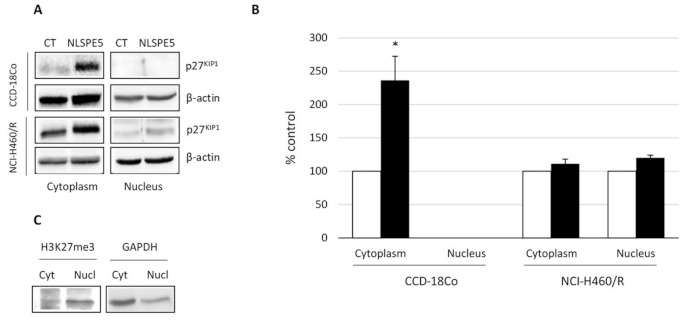
Western blot analysis of the accumulation of p27^KIP1^ in the nucleus and cytosol of CCD-18Co and NCI-H460/R. (**A**) Representative Western blots. (**B**) Expression levels relative to that of actin. Values of untreated cells are considered as 100%. (**C**) Western blots to control cross-contamination between nuclear and cytosolic samples. It is shown the expression of histone H3 and GADPH in the nucleus and cytosol, respectively. Differences versus untreated control cells were considered significant at * *p* < 0.05.

**Table 1 molecules-26-01319-t001:** IC50 (µM) values of NLSPE5 and onconase for a panel of tumor and non-tumor cell lines.

	Cell Line	Origin	ONC IC_50_ (μM)	NLSPE5 IC_50_ (μM)
**Tumor Cell Lines**	NCI-H460	Large cell lung carcinoma	0.40 ± 0.06	0.15 ± 0.05
	OVCAR-8	Ovarian serous adenocarcinoma	0.54 ± 0.08	0.24 ± 0.04
	NCI-H460/R	NCI-H460 doxorubicin resistant	0.50 ± 0.10	0.40 ± 0.10
	Sk-Br-3	Breast cancer derived from metastatic pleural effusion	4.20 ± 0.60	0.36 ± 0.09
	HeLa	Cervix carcinoma	0.15 ± 0.02	0.26 ± 0.04
**Non-Tumor Cell Lines**	HaCaT	Spontaneously transformed keratinocyte	0.70 ± 0.09	0.92 ± 0.14
	CCD-18Co	Colon fibroblasts	0.56 ± 0.10	7.10 ± 1.04
	HEK-293	Embryonic kidney cells	0.28 ± 0.05	0.74 ± 0.18
	1BR3G	Skin transformed fibroblasts	0.50 ± 0.13	0.75 ± 0.10
	MCF10A	Epithelial mammary gland	1.07 ± 0.07	1.50 ± 0.09

**Table 2 molecules-26-01319-t002:** Effect of NLSPE5 on the cell cycle phase distribution of CCD-18Co and NCI-H460/R cells. Values were analyzed from 10,000 total events from three independent experiments.

	CCD-18Co		NCI-H460/R	
	Control	NLSPE5	Control	NLSPE5
**G_0_/G_1_**	74.7 ± 1.0	70.8 ± 2.8	61.2 ± 6.5	26.9 ± 5.2
**S**	15.4 ± 0.3	11.7 ± 0.4	20.4 ± 3.5	33.4 ± 2.7
**G_2_/M**	5.1 ± 0.4	6.4 ± 0.5	10.7 ± 2.2	16.0 ± 2.2
**subG_1_**	1.4 ± 0.5	10.6 ± 3.0	3.7 ± 0.8	18.9 ± 6.2

## Data Availability

The data presented are contained within the article or Appendix A.

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
