# Peer review of "The Selectivity for Tumor Cells of Nuclear-Directed Cytotoxic RNases Is Mediated by the Nuclear/Cytoplasmic Distribution of p27^KIP1^"

_molecules, 2021, doi:10.3390/molecules26051319_

Round 1

Reviewer 1 Report

The main messages of this manuscript are that ND-RNases are cancer selective cytotoxic; that they do not affect proliferation of non-tumor cells; and that these effects are related to p21 and p27 expression and cytosolic localization. The authors are asked to address the following issues:

Section 2.1:

To interpret the cancer cell selectivity of the test compound NLSPE5, the biology of the cell lines used in the in vitro experiments is important. The authors should discuss the properties of the non-malignant cells used. These appear mainly transformed cell lines, including an adenovirus oncogene-transformed cell line. Which of the non-tumor cell lines they used most accurately represent non-malignant control cells? Cytotoxic/cytostatic effects of compounds could depend on cell cycle. Therefore, what is the proliferation activity of the non-malignant control cell lines under the conditions used; and how does this compare to the cancer cell lines tested?

IC50 data are shown in tabular form only. The reliability of IC50 calculations from dose-response analyses is highly dependent on the shape of the sigmoidal dose-response curves, in particular on the normalization method used and the steepness of the slopes, which may differ considerably between cell lines. For proper interpretation of the results it is therefore essential that the authors show the actual dose-response curves, if not in the main manuscript then at least in a supplemental data file.

Section 2.2:

Here, the authors wish to investigate if the effects of NLSPE5 are cytostatic or cytotoxic. To answer this question they measure cell numbers and viability of the most resistant cancer cell line and most resistant non-malignant cell line, subjected to drug concentrations below the IC50. This seems an odd choice. Why not use cells on which the effects of NLSPE5 are more clear, using an effective dose? Now effects that do exist might be undetectable.

In the cell proliferation experiment (total cell count), all data appear normalized by the total cell number at each day. Due to this way of presenting the data it is not possible to assess if cells proliferated or not. Proliferation rates of the control cultures can be different for the two cell lines. From the data shown, readers can not conclude if 20% proliferation inhibition means only 20% reduction in cell numbers of a population that doubled every day; or complete arrest of cultures that expanded by only 20%. These two possibilities lead to different conclusions. Therefore, all data should be normalized by the total cell number at t=0.

Line 86-88 is unclear. How can we see the effects on non-tumor cells by looking at the data of the cancer cells?

Line 89-90: “NLSPE5 does not seem to affect the proliferation of CCD-18Co cells being the decrease of total cells equal to that of viable cells” and lines 94-95 “on tumor cells, the ND-RNase clearly decreases both their viability and proliferation.“ Such conclusions cannot be drawn simply by comparing the proportion viable cells to the relative total number of cells on a given day. Cells that die do not proliferate anymore and thus do not contribute to all subsequent cell doublings anymore.

Table 2: Cell cycle phase analysis by flow cytometry is quite dependent on the settings of thresholds in the analysis software. At least typical examples of the DNA histograms should be shown.

Lines 98-104: The occurrence of a substantial fraction of treated cells in sub-G1, i.e. probably dead cells, should not be neglected.

Section 2.3:

Lines 109-115: Apoptosis induction is measured only on CCD-18Co cells. Apoptosis is clearly induced. The result is compared to a historic analysis on cancer cell line NCI/ADR-RES, where apoptosis induction was apparently stronger. This indirect comparison is then used to explain the difference in toxicity observed on CCD-18Co cells versus NCI-H460/R cells. In particular given the significant differences in toxicity observed on different cell lines, such an extrapolation of the results should be avoided. The apoptosis analysis should at least be done on NCI-H460/R cells as well.

Figures 3 and 4 and lines 122-133: An induction in p21 and p27 protein levels in non-malignant cells is shown in support of the statements that these CDKIs “display an important role on the higher resistance of non-tumor cells to these ribonucleases”(lines 27-28) and that “that the higher resistance of CCD-18Co cells to the RNase-induced apoptosis relies on the overexpression and cytoplasmic subcellular localization of p21WAF1/CIP1and p27KIP1” (lines 165-166). To allow drawing these conclusions, the same analysis should be done on a cancer cell line, preferably NCI-H460/R cells, to show that in these cells p21 and p27 levels are not increased in the cytoplasm. An indirect historic comparison to p27 levels in different cancer cell lines (in part treated with a different ND-RNase) cannot fully substitute for this omission.

Figure 4: To show that this Western blot shows cytosolic proteins only, control hybridizations with antibodies detecting paradigm nucleic and cytosolic proteins should be presented.

Specific points:

References 12 and 18 are identical. References 15-18 are cited in the text after reference 29.

Line 357: Samples of the ND-RNases are not available from the authors. This conflicts with research ethics. The ND-RNases are the compounds under study here. Not making the compounds available to the scientific community precludes independent confirmation of the reported findings, which is not acceptable.

Author Response

February 17th, 2021

Dear Sirs, 

Please find enclosed a revised version of our manuscript entitled “The Selectivity for Tumor Cells of Nuclear-Directed Cytotoxic RNases Is Mediated by the Nuclear/Cytoplasmic Distribution of p27KIP1” (G. García-Galindo, J. Castro, J. Matés, M. Bravo, M. Ribó, M. Vilanova and A. Benito, Manuscript ID: molecules-996291) which we hereby submit for publication in Molecules. We would like to thank the reviewers for their time and attention in reviewing our manuscript and for their constructive comments. We have carried out the experiments suggested by them, which have implicated a new researcher (Mr. Marlon Bravo) included now as new author. We believe that the new results have contributed to improve the work. From the experiments carried out, the conclusions of the manuscript are different to those of the former version and the title has changed accordingly. All the changes introduced in the manuscript are marked in red. The answers to the comments of the reviewers and the actions we have taken are detailed below. We hope that now the manuscript will be found acceptable for publication in Molecules.

Sincerely,

Antoni Benito

Maria Vilanova

REVIEWER 1

We are grateful to the reviewer for their comments and constructive suggestions. We have amended our manuscript according to them as follows:

To interpret the cancer cell selectivity of the test compound NLSPE5, the biology of the cell lines used in the in vitro experiments is important. The authors should discuss the properties of the non-malignant cells used. These appear mainly transformed cell lines, including an adenovirus oncogene-transformed cell line. Which of the non-tumor cell lines they used most accurately represent non-malignant control cells? Cytotoxic/cytostatic effects of compounds could depend on cell cycle. Therefore, what is the proliferation activity of the non-malignant control cell lines under the conditions used; and how does this compare to the cancer cell lines tested?

The following discussion of the different non-tumor cells used in this work has been included in page 6, lines 178-194:

“We tested the effect of NLSPE5 on different non-tumor cells lines. HaCat cell line is aneuploid but, despite exhibiting phenotypic traits of transformation in vitro, is non tumorigenic, closely resemble normal human keratinocytes in their growth and differentiation potential and expresses a virtually normal pattern of differentiation [18-20]. CCD-18Co cells are primary normal human colon fibroblasts and have been extensively used as a healthy non-stem-cell and non-cancer-cell in vitro reference model [21-24]. MFC10A cells are derived from benign proliferative breast tissue and spontaneously immortalized without defined factors. They are not tumorigenic and do not express estrogen receptor. Furthermore, when cultured on top of MatrigelTM, MCF10A cells are capable of forming acinus-like spheroids with a hollow lumen [25]. This structure is covered by basement membrane and formed by polarized and organized cells [26]. Regarding 1BR3G cells, they are SV40-immortalized skin fibroblasts derived from a phenotypically normal individual with a normal capacity for DNA DSB repair. They have also been previously used as normal cells [27-29]. Finally, HEK-293 cells were generated by transformation of culture of normal human embryonic kidney cells with sheared human adenovirus type 5 DNA [30]. Although these cells are immortalized by a known oncogene, they are not malignant [31] and have previously been used as a non-tumor cell model [32,33]. Interestingly, among these cell models, the best established as a cell model of normal cells (CCD-18Co) is the less affected by NLSPE5.”

Since we wanted to investigate the causes of the selectivity of the ND-RNase, we compared healthy normal cells with a tumor cell line representative of poor prognostic cancer and therefore proliferation rates were necessarily different. We cannot discard that we have not detected a cytostatic effect of NLSPE5 on non-tumor cells due to their long doubling time. Therefore the text has been changed as follows: “In contrast, NLSPE5 does not seem to affect the proliferation of CCD-18Co cells, at least at the incubation times investigated, being the decrease of total cells equal to that of viable cells. It is worth mentioning that the effect of NLSPE5 on the NCI-H460/R cell line is very similar to that previously observed for another ND-RNase named PE5 in the ovarian MDR cell line NCI/ADR-RES [12] but for NCI-H460/R cells the effect on proliferation is more evident. In summary, NLSPE5 has a mince effect on the viability of non-tumor cells and likely does not affect their proliferation although, on tumor cells, the ND-RNase decreases both their viability and proliferation.” (lines 90-97 page 3). Nevertheless, the hypothesis of a lack of effect on the proliferation rate seems to be confirmed on the cell cycle assay with CCD-18Co cells. In this cell line NLSPE5 does not significantly alter the proportion of cell cycle phases likely indicating that cells have not been arrested at any phase.

IC50 data are shown in tabular form only. The reliability of IC50 calculations from dose-response analyses is highly dependent on the shape of the sigmoidal dose-response curves, in particular on the normalization method used and the steepness of the slopes, which may differ considerably between cell lines. For proper interpretation of the results it is therefore essential that the authors show the actual dose-response curves, if not in the main manuscript then at least in a supplemental data file.

According to the reviewer’s comment, we have included the dose-response curves as supplementary material in the new version (Figure S1 of Supplementary Material).

Section 2.2:

Here, the authors wish to investigate if the effects of NLSPE5 are cytostatic or cytotoxic. To answer this question they measure cell numbers and viability of the most resistant cancer cell line and most resistant non-malignant cell line, subjected to drug concentrations below the IC50. This seems an odd choice. Why not use cells on which the effects of NLSPE5 are more clear, using an effective dose? Now effects that do exist might be undetectable.

In this work we have investigated about the differential effect of NLSPE5 between normal and tumor cells. As normal cells we have chosen CCD-18Co because they are those cells that most accurately represent non-malignant cells among the assayed panel. On the contrary, NCI-H460/R is a cell line representative of poor prognostic small cell lung cancer and is resistant to doxorubicin (96.2-fold) and cross-resistant to etoposide, paclitaxel, vinblastine and epirubicin (Pesic et al., 2006 Chemother.18(1):66-73). Interestingly, non-tumor CCD-18Co cells are those most resistant to NLSPE5 and NCI-H460/R cell line is one of the most sensitive. Therefore, these two models allow us to investigate why NLSPE5 affects much more cancer cells than normal cells. Finally, we must stress out that for experiments comparing the effects of NLSPE5 between both cell lines we have not used the same concentration of drug but their isoeffect concentrations (i.e. the corresponding IC50 in each case) discarding that the effect could be undetectable.

In the cell proliferation experiment (total cell count), all data appear normalized by the total cell number at each day. Due to this way of presenting the data it is not possible to assess if cells proliferated or not. Proliferation rates of the control cultures can be different for the two cell lines. From the data shown, readers can not conclude if 20% proliferation inhibition means only 20% reduction in cell numbers of a population that doubled every day; or complete arrest of cultures that expanded by only 20%. These two possibilities lead to different conclusions. Therefore, all data should be normalized by the total cell number at t=0.

We agree with the referee that using this way of representing the effect of NLSPE5 on proliferation did not allow to assess whether the cells were proliferating or not. In the new version we have modified Figure 1 (page 3) by normalizing by the total cell number at t=0, as suggested.

Line 86-88 is unclear. How can we see the effects on non-tumor cells by looking at the data of the cancer cells?

We agree with the reviewer that the sentence was confusing. Now it says: “When comparing both cell lines we can see that the effect of NLSPE5 on cell viability is much lower in non-tumor cells since to achieve the same rate of viable cells we have to use much higher concentrations of the ND-RNase.” (lines 87-89)

Line 89-90: “NLSPE5 does not seem to affect the proliferation of CCD-18Co cells being the decrease of total cells equal to that of viable cells” and lines 94-95 “on tumor cells, the ND-RNase clearly decreases both their viability and proliferation.“ Such conclusions cannot be drawn simply by comparing the proportion viable cells to the relative total number of cells on a given day. Cells that die do not proliferate anymore and thus do not contribute to all subsequent cell doublings anymore.

Regarding the sentence between lines 89-90 of the former version, we agree with the reviewer that death cells do not proliferate and therefore the appearance of death cells imply that the amount of total cells should decrease. However, due to this fact it is expected that if the drug is also decreasing the proliferation rate the decrease of percentage of total cells in the treated cultures should be higher than the decrease of percentage of live cells. In our case, the decrease of the percentages of live and total cells are parallel in NLSPE5 treated cultures implicating that the effect of the ND-RNase on proliferation in this case is negligible. On the contrary, regarding tumor cells, we observe that at the lower concentrations of NLSPE5 a decrease of 10% in viability is accompanied by a 300%-fold decrease in proliferation. Although part of the decrease of total cells could be due to the presence of death cells, we believe that our results indicate that NLSPE5 is also producing a cytostatic effect on the NCI-H460/R cell line. Nevertheless, since the proliferation rates of both cell models are quite different, although cell cycle analysis experiments corroborate our hypothesis we have rewritten the paragraph as follows:

“When comparing both cell lines we can see that the effect of NLSPE5 on cell viability is much lower in non-tumor cells since to achieve the same rate of viable cells we have to use much higher concentrations of the ND-RNase. NLSPE5 reduces the cell proliferation of NCI-H460/R cells even at 24h of incubation and this effect increases along the time. In contrast, NLSPE5 does not seem to affect the proliferation of CCD-18Co cells, at least at the incubation times investigated, being the decrease of total cells equal to that of viable cells. It is worth mentioning that the effect of NLSPE5 on the NCI-H460/R cell line is very similar to that previously observed for another ND-RNase named PE5 in the ovarian MDR cell line NCI/ADR-RES [12] but for NCI-H460/R cells the effect on proliferation is more evident. In summary, NLSPE5 has a mince effect on the viability of non-tumor cells and likely does not affect their proliferation although, on tumor cells, the ND-RNase decreases both their viability and proliferation.” (Lines 87-97, page 3).

Table 2: Cell cycle phase analysis by flow cytometry is quite dependent on the settings of thresholds in the analysis software. At least typical examples of the DNA histograms should be shown.

We have included examples of the flow cytometry DNA histograms as supplementary material in the new version (Figure S2 of Supplementary Material).

Lines 98-104: The occurrence of a substantial fraction of treated cells in sub-G1, i.e. probably dead cells, should not be neglected.

We agree with the reviewer that the fraction of treated cells in sub-G1 is significant and this has been noted now in the text (page 3). In lines 108-110 it is stated that “A significant fraction of the cells are on sub-G1, which is indicative that these cells have entered on apoptosis. This effect is higher on tumor cells (Table 2)”.

Section 2.3:

Lines 109-115: Apoptosis induction is measured only on CCD-18Co cells. Apoptosis is clearly induced. The result is compared to a historic analysis on cancer cell line NCI/ADR-RES, where apoptosis induction was apparently stronger. This indirect comparison is then used to explain the difference in toxicity observed on CCD-18Co cells versus NCI-H460/R cells. In particular given the significant differences in toxicity observed on different cell lines, such an extrapolation of the results should be avoided. The apoptosis analysis should at least be done on NCI-H460/R cells as well.

The objective of this work was to assess the effect of NLSPE5 on non-tumor cells. Our sentence was an extrapolation to compare with previous results of our group and may be the “Results” section is not the more appropriate part of the manuscript to include it. This extrapolation has been therefore moved to the “Discussion” section (lines 195-198, page 6).  

Figures 3 and 4 and lines 122-133: An induction in p21 and p27 protein levels in non-malignant cells is shown in support of the statements that these CDKIs “display an important role on the higher resistance of non-tumor cells to these ribonucleases”(lines 27-28) and that “that the higher resistance of CCD-18Co cells to the RNase-induced apoptosis relies on the overexpression and cytoplasmic subcellular localization of p21WAF1/CIP1and p27KIP1” (lines 165-166). To allow drawing these conclusions, the same analysis should be done on a cancer cell line, preferably NCI-H460/R cells, to show that in these cells p21 and p27 levels are not increased in the cytoplasm. An indirect historic comparison to p27 levels in different cancer cell lines (in part treated with a different ND-RNase) cannot fully substitute for this omission.

We agree with the reviewer that this is an important point. To be sure that p21 and p27 display an important role on the higher resistance of non-tumor cells to the ND-RNases we have had to demonstrate that this effect is not present on tumor cells. Accordingly, we carried out this experiment with the NCI-H460/R cells and we checked again the amount of p27KIP1 and p21WAF1/CIP1 on the nucleus and cytoplasm upon treatment with NLSPE5. Our experiments show that the nuclear and cytoplasmic amounts of p27KIP1 in NCI-H460/R do not change upon treatment with NLSPE5. These results corroborate that the overexpression and cytoplasmic subcellular localization of p27KIP1 display an important role on the higher resistance of non-tumor cells to NLSPE5, On the contrary, we observe that the amount of cytoplasmic p21WAF1/CIP1 in NCI-H460/R increases significantly when the cells are treated with the ND-RNase. This experiment indicates that our conclusion that p21WAF1/CIP1 is important for the resistance of non-tumor cells to NLSPE5 was not valid and therefore we have suppressed it together with the related experiments from the present version of the manuscript. We thank the reviewer for this critical comment since it has avoided that we report a non-correct conclusion.

Figure 4: To show that this Western blot shows cytosolic proteins only, control hybridizations with antibodies detecting paradigm nucleic and cytosolic proteins should be presented.

We have checked that the cytoplasmic fraction does not contain part of the nuclear fraction by analyzing it in a Western blot using antibodies against histone H3 (H3K27me3). Conversely, we have checked that the nuclear fraction does not contain part of the cytoplasmic fraction by analyzing it in a Western blot using antibodies against GAPDH. As it can be now seen in Figure 4, the main fraction of histone H3 is detected in the nuclear compartment and that of GAPDH on the cytoplasmic compartment. The fainter bands observed in the other compartment account for around 5% of the intensity for histone H3 and 20% of the intensity for GAPDH, and therefore do not alter the observations described of the manuscript.

Specific points:

References 12 and 18 are identical. References 15-18 are cited in the text after reference 29.

We apologize for the error. We have checked throughout the references of the manuscript.

Line 357: Samples of the ND-RNases are not available from the authors. This conflicts with research ethics. The ND-RNases are the compounds under study here. Not making the compounds available to the scientific community precludes independent confirmation of the reported findings, which is not acceptable.

As it is now stated, we have decided to make available samples of NLSPE5 to the scientific community.

Reviewer 2 Report

Major comments

In this study, entitled “The Selectivity for Tumor Cells of Nuclear-Directed Cytotoxic RNases Is Mediated by the Nuclear/Cytoplasmic Distribution of p27KIP1 and p21WAF1/CIP1”, authors explored the cytotoxic effects of an engineered ribonuclease NLSPE5 on cancer cells. Their findings showed that NLSPE5 had higher IC50 ratios between non-tumor cells and tumor cells, and clearly induced cell cycle arrest at S and G2/M phase and activation of caspase-3, -8, and -9. Further investigation showed that NLSPE5 induced the expression of cyclin D1, cyclin E, p21 and p27. Generally, this study has interest and merit, most experiments have been properly performed, and most conclusion are supported by the results. Several concerns are needed to be further clarified and interpreted.

  1. In Figure 1, are the differences of cell viability between control and treatment significant? Similarly, in Figure 2, are the differences of caspase-3, -8, and -9 between control and treatment significant? Authors should conduct proper statistical analysis for all the quantitative data in figures and tables.
  2. In Figure 3, the images of western blot seem to be picked and pasted. Authors should provide the full image of original data. In addition, the most influenced proteins such as p21, p27, cyclin D1, and cyclin E, are the downstream targets of p53. Thus, authors should evaluate the effect of NLSPE5 on p53.
  3. Authors only used a single dose of NLSPE5 to evaluate the anticancer effects of NLSPE5 in the study. It is suggested to determine whether the anticancer effects of NLSPE5 is dose-dependent or not.
  4. There are many confusing symbols in the whole manuscript. Authors should well revise them.

Author Response

February 17th, 2021

Dear Sirs, 

Please find enclosed a revised version of our manuscript entitled “The Selectivity for Tumor Cells of Nuclear-Directed Cytotoxic RNases Is Mediated by the Nuclear/Cytoplasmic Distribution of p27KIP1” (G. García-Galindo, J. Castro, J. Matés, M. Bravo, M. Ribó, M. Vilanova and A. Benito, Manuscript ID: molecules-996291) which we hereby submit for publication in Molecules. We would like to thank the reviewers for their time and attention in reviewing our manuscript and for their constructive comments. We have carried out the experiments suggested by them, which have implicated a new researcher (Mr. Marlon Bravo) included now as new author. We believe that the new results have contributed to improve the work. From the experiments carried out, the conclusions of the manuscript are different to those of the former version and the title has changed accordingly. All the changes introduced in the manuscript are marked in red. The answers to the comments of the reviewers and the actions we have taken are detailed below. We hope that now the manuscript will be found acceptable for publication in Molecules.

Sincerely,

Antoni Benito

Maria Vilanova

REVIEWER 2

We are grateful to the reviewer for their comments and constructive suggestions. We have amended our manuscript according to them as follows:

In Figure 1, are the differences of cell viability between control and treatment significant? Similarly, in Figure 2, are the differences of caspase-3, -8, and -9 between control and treatment significant? Authors should conduct proper statistical analysis for all the quantitative data in figures and tables.

Regarding Figure 1 we have not carried out the statistical assay, however, taking into account the error bars obtained for each of the measured IC50, we believe that the general trends described in the text are correct. Regarding Figure 2, we have performed the statistic analysis and

Figure 2 (page 4) now indicates that the degree of activation of the three procaspases is significantly different respective to untreated control cells.

In Figure 3, the images of western blot seem to be picked and pasted. Authors should provide the full image of original data. In addition, the most influenced proteins such as p21, p27, cyclin D1, and cyclin E, are the downstream targets of p53. Thus, authors should evaluate the effect of NLSPE5 on p53.

We do not have full images of the immobilon sheets because in order to economize we assay different antibodies in the same membrane by cutting it using molecular mass standards as references.

The fact that p21WAF1/CIP1, p27KIP1, cyclin D1, and cyclin E are downstream targets of p53 is an interesting observation. However, when we compare the p53 status of the panel of cells used in this study we do not see any correlation between p53 activity and sensitivity to NLSPE5. As examples among the non-tumor cells, both the most resistant CCD-18Co and the most sensitive to NLSPE5 HEK-293 are wt p53 (Hayashi et al 2016 Carcinogenesis 37 (10) 972-984; Sasaki et al 2011 Carcinogenesis 37 (10) 972-984). Regarding tumor cells NCI-H460 has wt p53 (Lai et al., 2000 J Biomed Sci 7(1): 64-70 and is the most sensitive cell line to NLSPE5. This is now stated in the Discussion between lines 199-201: “p21WAF1/CIP1, p27KIP1, cyclin D1, and cyclin E are the downstream targets of p53 but we have not found any correlation between the status of p53 and their sensitivity to NLSPE5 among the panel of investigated cell lines.”

Authors only used a single dose of NLSPE5 to evaluate the anticancer effects of NLSPE5 in the study. It is suggested to determine whether the anticancer effects of NLSPE5 is dose-dependent or not.

Each IC50 value indicated in Table 1 is obtained from an experiment in which the effect of six different concentrations is measured. In each case, when representing the concentration of the incubated drug vs the observed effect in the cell line, we obtain sigmoidal curves from which we calculate the IC50 value. Therefore, in all cases the effect is dose-dependent. In the new version we have included the different dose-response curves as supplementary material (Figure S1 of Supplementary Material).

There are many confusing symbols in the whole manuscript. Authors should well revise them.

We apologize for the confusing symbols. We have checked throughout the Figures and we believe that now the symbols are clear.

Round 2

Reviewer 1 Report

Most issues were resolved.

However, the newly added supplementary figure 2 has raised a small concern that should be corrected. The insets show the mean data of three independent analyses (same data as shown in table 2). This should be replaced by the actual data from the experiment shown. Otherwise, this cannot be considered a representative example of the three experiments done.

Further, in my opinion the text “relies on” on line 202 should be changed. This suggests that the authors have obtained formal proof for the mechanistic link between NLSPE5 expression and protection from cell death. This is not the case. They have shown that both occur, but not that p27 cytoplasmic localization protects CCD-18co cells from apoptosis. This would have required some sort of rescue experiment, such as silencing p27. Therefore, “coincides with” or something alike would be more appropriate. On line 207, the authors state that p27 cytoplasmic localization “may explain” resistance from apoptosis, which is accurate.

Finally, while the text is understandable, editing is recommended to correct the many grammatical errors.

Author Response

Dear Sirs, 

Please find enclosed a revised version (2nd round) of our manuscript entitled “The Selectivity for Tumor Cells of Nuclear-Directed Cytotoxic RNases Is Mediated by the Nuclear/Cytoplasmic Distribution of p27KIP1” (G. García-Galindo, J. Castro, J. Matés, M. Bravo, M. Ribó, M. Vilanova and A. Benito, Manuscript ID: molecules-996291) which we hereby submit for publication in Molecules. We would like to thank the reviewers for their time and attention in reviewing our manuscript and for their constructive comments. The answers to the minor comments of the reviewers and the actions we have taken are detailed below. We hope that now the manuscript will be found acceptable for publication in Molecules.

Sincerely,

Antoni Benito

Maria Vilanova

REVIEWER 1

We are grateful to the reviewer for their comments and constructive suggestions. We have amended our manuscript according to them as follows:

Most issues were resolved. However, the newly added supplementary figure 2 has raised a small concern that should be corrected. The insets show the mean data of three independent analyses (same data as shown in table 2). This should be replaced by the actual data from the experiment shown. Otherwise, this cannot be considered a representative example of the three experiments done.

Accordingly to the reviewer, we have replaced the mean values in the insets of supplementary figure S2 by those of the corresponding to the actual experiment shown there.

Further, in my opinion the text “relies on” on line 202 should be changed. This suggests that the authors have obtained formal proof for the mechanistic link between NLSPE5 expression and protection from cell death. This is not the case. They have shown that both occur, but not that p27 cytoplasmic localization protects CCD-18co cells from apoptosis. This would have required some sort of rescue experiment, such as silencing p27. Therefore, “coincides with” or something alike would be more appropriate. On line 207, the authors state that p27 cytoplasmic localization “may explain” resistance from apoptosis, which is accurate.

We have made the change accordingly. Now, the sentence says: “Here, we show that NLSPE5 affects specifically tumor cells in vitro and that the higher resistance of CCD-18Co cells to the RNase-induced apoptosis coincides with the overexpression and cytoplasmic subcellular localization of p27KIP1.” (lines 205-207 of the new version)

Finally, while the text is understandable, editing is recommended to correct the many grammatical errors.

The manuscript has been revised by the Modern Language Facility of our University.

Reviewer 2 Report

The revised manuscript has been significantly improved, and most previous concerns are also solved. However, the statistical significance of changes in the quantitative bar plots in Figure 3 and Figure 4 are still unclear. Authors should perform proper statistical analysis for them and clearly indicate them.

Author Response

Dear Sirs, 

Please find enclosed a revised version (2nd round) of our manuscript entitled “The Selectivity for Tumor Cells of Nuclear-Directed Cytotoxic RNases Is Mediated by the Nuclear/Cytoplasmic Distribution of p27KIP1” (G. García-Galindo, J. Castro, J. Matés, M. Bravo, M. Ribó, M. Vilanova and A. Benito, Manuscript ID: molecules-996291) which we hereby submit for publication in Molecules. We would like to thank the reviewers for their time and attention in reviewing our manuscript and for their constructive comments. The answers to the minor comments of the reviewers and the actions we have taken are detailed below. We hope that now the manuscript will be found acceptable for publication in Molecules.

Sincerely,

Antoni Benito

Maria Vilanova

REVIEWER 2

We are grateful to the reviewer for their comments and constructive suggestions. We have amended our manuscript according to them as follows:

 The revised manuscript has been significantly improved, and most previous concerns are also solved. However, the statistical significance of changes in the quantitative bar plots in Figure 3 and Figure 4 are still unclear. Authors should perform proper statistical analysis for them and clearly indicate them.

We have carried out the statistical analysis of the results presented in Figures 3 and 4. The significant statistical differences between control and treated cell cultures are now marked with an asterisk. The analysis of the results of Figure 3 shows that the increase of expression of cyclin D1 is not significant but this does not change the conclusions of the manuscript. According to this analysis, the text now says: “A significant increase is observed for p21WAF1/CIP1, p27KIP1 and cyclin E” (line 133).